# Structural and Functional Characteristics of TLR19 in Barbel Chub Compared to TLR19 in Grass Carp

**DOI:** 10.3390/ijms26073103

**Published:** 2025-03-27

**Authors:** Zhao Lv, Mengyuan Zhang, Yang Xu, Beibei Qin, Hong Yang, Ruizhong Wei, Tiaoyi Xiao

**Affiliations:** Hunan Engineering Technology Research Center of Featured Aquatic Resources Utilization, Hunan Agricultural University, Changsha 410128, China; zhangmengyuanlh@163.com (M.Z.); xy_961112@163.com (Y.X.); qinbeibei0@gmail.com (B.Q.); hongyang1920@163.com (H.Y.); 17689745640@163.com (R.W.); tiaoyixiao@hunau.edu.cn (T.X.)

**Keywords:** barbel chub, grass carp, TLR19, structure, immune function

## Abstract

The hybrid offspring of barbel chub *Squaliobarbus curriculus* and grass carp *Ctenopharyngodon idella* exhibit stronger resistance to the grass carp reovirus (GCRV) infection than grass carp. Toll-like receptors (TLRs) play indispensable roles in the antiviral immunity of fish. In this study, the structures and antiviral immune functions of barbel chub TLR19 (ScTLR19) and grass carp TLR19 (CiTLR19) were compared. The amino acid sequence of ScTLR19 shared high similarity (97.4%) and identity (94.0%) with that of CiTLR19, and a phylogenetic tree revealed the close evolutionary relationship between ScTLR19 and CiTLR19. Protein domain composition analyses showed that ScTLR19 possessed an additional leucine-rich repeat (designated as LRR9) located at amino acid positions 654–677 in the extracellular region, which was absent in CiTLR19. Multiple sequence alignment and three-dimensional structure comparison also indicated that the extracellular regions of ScTLR19 and CiTLR19 exhibited greater differences compared to their intracellular regions. Molecular docking revealed that the extracellular region of ScTLR19 (docking score = −512.31) showed a stronger tendency for binding with polyI:C, compared to the extracellular region of CiTLR19 (docking score = −474.90). Replacing LRR9 in ScTLR19 with the corresponding amino acid sequence from CiTLR19 reduced the binding activity of ScTLR19 to polyI:C, as confirmed by an ELISA. Moreover, overexpression experiments suggested that ScTLR19 could regulate both the IRF3–TRIF and IRF3–MyD88 signaling pathways during GCRV infection, while CiTLR19 only regulated the IRF3–MyD88 signaling pathway. Importantly, replacing LRR9 in ScTLR19 with the corresponding amino acid sequence from CiTLR19 altered the expression regulation on IRF3, MyD88, and TRIF during GCRV infection. These findings collectively reveal the structural and functional differences between ScTLR19 and CiTLR19, and they may provide data to support a deeper understanding of the molecular mechanisms underlying the differences in GCRV resistance between barbel chub and grass carp, as well as the genetic basis for the heterosis of GCRV resistance in their hybrid offspring.

## 1. Introduction

Fish, as one of the most primitive vertebrates, possess a relatively less developed acquired immune system compared to that of higher vertebrates including mammals, primarily relying on the innate immune system to defend against pathogen invasion [1,2]. It is well established that the innate immune system depends on pattern recognition receptors (PRRs) to recognize pathogen-associated molecular patterns, subsequently activating downstream signaling pathways and inducing immune responses to eliminate or clear invading pathogens [3,4]. The toll-like receptor (TLR) family is one of the most extensively studied PRRs to date [5]. Based on the ligand categories and structural characteristics of mammalian TLRs, over 20 identified TLRs in fish have been classified into six major subfamilies, namely the TLR1, TLR3–5, TLR7, and TLR11 subfamilies [6,7,8]. The basic structure of TLRs includes a transmembrane region, an extracellular region, and an intracellular region [9]. All TLRs possess a single transmembrane domain, classifying them as type I transmembrane proteins [10]. The extracellular region of TLRs is composed of multiple leucine-rich repeat (LRR) motifs, and the number of LRRs varies among different TLRs, leading to structural and ligand specificity [11]. The intracellular region of TLRs typically consists of a toll/interleukin-1 receptor (TIR) domain, which is a critical structure for TLR signal transduction [12]. This domain can bind and activate intracellular adapter proteins, cascading the activation of downstream immune signals [12]. The signaling cascades mediated by different TLRs are primarily divided into myeloid differentiation factor 88 (MyD88)-dependent and TIR domain-containing adaptor-inducing interferon-β (TRIF)-dependent pathways, ultimately activating nuclear transcription factors such as interferon regulatory factors (IRFs) and nuclear transcription factor-κB (NF-κB), and inducing the expression of interferons, interleukins, chemokines, and other immune mediators, thereby eliciting diverse immune responses in the hosts [13,14].

TLR19 is a fish-specific TLR that belongs to the TLR11 subfamily, along with TLR11–13, TLR20–23, and TLR26 [15,16,17,18]. Currently, functional studies on the TLR11 subfamily in fish have primarily focused on TLR22, which is highly expressed in most immune tissues of fish, suggesting that TLR22 may be an important immune gene in resisting pathogen invasion [19,20,21]. To date, the TLR19 genes have been identified in various teleost species, such as channel catfish *Ictalurus punctatus* [22], zebrafish *Danio rerio* [23], Atlantic salmon *Salmo salar* [24], yellow catfish *Tachysurus fulvidraco* [25], blunt snout bream *Megalobrama amblycephala* [26], and grass carp *Ctenopharyngodon idella* [16]. In grass carp, the challenge of double-stranded RNA (dsRNA) viral analog polyI:C significantly upregulates the expression of the TLR19 gene in grass carp kidney cells (CIKs) [16]. In contrast, an infection with infectious salmon anemia virus (ISAV) in Atlantic salmon leads to a downregulation of TLR19 gene expression in the kidney [24]. Furthermore, stimulation with inactivated *Aeromonas hydrophila* significantly downregulates TLR19 gene expression in the liver, spleen, and head kidney of blunt snout bream [26], while it significantly upregulates TLR19 gene expression in the liver, spleen, and blood of yellow catfish [25]. Therefore, TLR19 in different fish species exhibits distinct expression patterns in response to the same pathogen, implying that the functional roles and mechanisms of TLR19s may be species-specific.

Barbel chub *Squaliobarbus curriculus* belongs to the family Cyprinidae, characterized by strong adaptability, a rapid growth rate, and omnivorous feeding habits [27]. Notably, barbel chub exhibits greater resistance to grass carp reovirus (GCRV) infection compared to grass carp, which belongs to the same subfamily Leuciscinae as barbel chub and is the most widely cultured freshwater fish worldwide [28]. In our previous studies, artificial hybridization was conducted using barbel chub as the paternal parent and grass carp as the maternal parent, resulting in hybrid offspring with significantly enhanced resistance compared to grass carp [27]. Therefore, barbel chub serves as a donor of resistance gene resources for the hybrid offspring of barbel chub and grass carp [27]. However, the molecular mechanisms underlying the resistance differences between barbel chub and grass carp remain unclear, and the genetic and molecular basis for the heterosis of GCRV resistance in their hybrid offspring has yet to be fully elucidated. Given the critical role of PRRs in the antiviral immunity of fish, existing studies have suggested that certain TLRs and retinoic acid-induced gene-I like receptors (RLRs) in barbel chub exhibit significant structural differences compared to their counterparts in grass carp [29,30,31,32]. These differences may contribute to variations in their antiviral immune functions, thereby leading to the observed differences in GCRV resistance between these two fish.

The immune function of grass carp TLR19 (CiTLR19) has been extensively studied [16]. As an intracellular PRR, CiTLR19 recognizes the GCRV and activates the TRIF signaling pathway, leading to the activation of IRF3 and the induction of interferon-mediated antiviral responses, thereby exerting antiviral immune functions [16]. In this study, the full-length cDNA sequence of barbel chub *S. curriculus* TLR19 (ScTLR19) was cloned. Structurally, ScTLR19 was found to contain one additional leucine-rich repeat (LRR) in its extracellular region compared to CiTLR19. Further comparative analyses were conducted to investigate the functional characteristics of ScTLR19 in immune recognition and the activation of antiviral immune pathways relative to CiTLR19. This study aims to reveal the structural and functional differences between ScTLR19 and CiTLR19, providing data to support a deeper understanding of the molecular mechanisms underlying the differences in GCRV resistance between barbel chub and grass carp, as well as the genetic basis for the heterosis of GCRV resistance in their hybrid offspring.

## 2. Results

### 2.1. Sequence Similarity and Homology of TLR19s from Barbel Chub and Grass Carp

The full-length cDNA sequence of the ScTLR19 gene was obtained through the rapid amplification of cDNA end (RACE), measuring 3230 bp (GenBank: MN432880; Figure 1). It includes an 84 bp 5′ untranslated region (UTR) and a 275 bp 3′ UTR, with the 3′ UTR containing a stop codon (TGA) and a poly(A) tail signal (Figure 1). The open reading frame (ORF) of ScTLR19 spans 2871 bp, encoding 957 amino acids (Figure 1), which is consistent with that of CiTLR19 [16]. The Expasy online platform predicted the molecular weight of the ScTLR19 protein to be 109.528 kDa, with a theoretical isoelectric point of 5.99.

A similarity analysis was conducted on the amino acid sequences of TLR19s from nine fish species, including barbel chub and grass carp. The results revealed that ScTLR19 exhibited the highest similarity (97.4%) with CiTLR19, followed by TLR19 from blunt snout bream with a similarity of 91.7% (Figure 2A). The amino acid similarity of ScTLR19 with TLR19s from other fish species ranged between 68.2 and 90.0%, while the amino acid similarity between ScTLR19 and TLR11 from mouse *Mus musculus* was 46.6% (Figure 2A). Homology analysis of the amino acid sequences also demonstrated that ScTLR19 shared the highest identity (94.0%) with CiTLR19 (Figure 2A).

A phylogenetic tree was constructed by using the reported TLR19 from nine fish species, as well as all the TLRs from humans and mice (Figure 2B). The results showed that the selected nine fish TLR19s were clustered with mouse TLR11, forming a distinct branch belonging to the vertebrate TLR11 subfamily, while being clearly separated from human and mouse TLR1–10 (Figure 2B). Among these nine fish TLR19s, ScTLR19 was first clustered with CiTLR19 and then grouped with TLR19s from blunt snout bream, common carp, and zebrafish to form a Cypriniformes branch (Figure 2B). This Cypriniformes branch first formed a sister clade with the Siluriformes branch, which included TLR19s from yellow catfish, Vachell’s bagrid fish *Tachysurus vachelli*, and channel catfish, and finally was clustered with the Salmoniformes branch, represented by Atlantic salmon TLR19 (Figure 2B).

### 2.2. Protein Domain Composition of TLR19s from Barbel Chub and Grass Carp

To clarify the structural differences between ScTLR19 and CiTLR19, the protein domains of TLR19s from barbel chub, grass carp, common carp, channel catfish, zebrafish, and Atlantic salmon were analyzed, along with the similarity of the amino acid sequences within each domain. The results revealed that all the TLR19 proteins from six fish species contained an N-terminal signal peptide, as well as an extracellular region including LRRs, an intracellular region including a TIR domain, and a transmembrane domain (TM) (Figure 3).

A detailed analysis of the positions and composition of the extracellular and intracellular domains revealed that ScTLR19 contained nine LRRs and one TIR domain, whereas CiTLR19 harbored eight LRRs and one TIR domain (Figure 3). The intracellular TIR domains in both ScTLR19 and CiTLR19 were located between amino acid positions 803–948 (Figure 3). The first eight LRRs and the TIR domains in ScTLR19 and CiTLR19 shared identical amino acid lengths and corresponding positions (Figure 3). However, ScTLR19 uniquely possessed an additional ninth LRR (LRR9) located at amino acid positions 654–677, which was absent in CiTLR19 (Figure 3).

The multiple sequence alignment results indicated that the transmembrane domains of ScTLR19 and CiTLR19 exhibited 100% amino acid similarity (Figure 4). With regard to the C-terminal intracellular region, particularly the TIR domain, the amino acid similarity between ScTLR19 and CiTLR19 was 99.3% (Figure 4). However, in the N-terminal LRR region, the amino acid similarity between ScTLR19 and CiTLR19 was only 95.8% (Figure 4).

### 2.3. Three-Dimensional Structural Features of TLR19s from Barbel Chub and Grass Carp

Using the Swiss-Model server, the three-dimensional structures of ScTLR19 and CiTLR19 were predicted. The results showed that ScTLR19 consisted of 39 β-sheets and 32 α-helices (Figure 5A), while CiTLR19 was composed of 39 β-sheets and 30 α-helices (Figure 5B). The structural alignment of the two TLR19 models using PyMol software 3.0 yielded a root-mean-square deviation (RMSD) of 0.614 (Figure 5C).

Furthermore, a detailed comparison of the extracellular and intracellular regions of ScTLR19 and CiTLR19 was conducted. Overall, the extracellular regions of both ScTLR19 (Figure 5D) and CiTLR19 (Figure 5E) exhibited a typical horseshoe-like structure, composed of 35 β-sheets and 21 α-helices. The structural alignment of the ScTLR19 and CiTLR19 extracellular regions using PyMol software resulted in an RMSD of 0.466 (Figure 5F). In contrast, the intracellular region of ScTLR19 consisted of four β-sheets and ten α-helices (Figure 5G), while that of CiTLR19 consisted of four β-sheets and eight α-helices (Figure 5H). However, the structural alignment of the ScTLR19 and CiTLR19 intracellular regions using PyMol software yielded an RMSD of 0.165 (Figure 5I), indicating that the intracellular regions of ScTLR19 and CiTLR19 exhibited smaller structural differences compared to their extracellular regions.

### 2.4. Differences in Binding Activity of TLR19 from Barbel Chub and Grass Carp to PolyI:C

To investigate whether there are differences in the binding activity of ScTLR19 and CiTLR19 to polyI:C, we performed molecular docking using the HDOCK server to analyze the interactions between the three-dimensional structures of the extracellular regions of ScTLR19 and CiTLR19 with polyI:C. The results revealed that polyI:C was positioned within the horseshoe-shaped cavity of the extracellular regions of both ScTLR19 and CiTLR19 (Figure 6). Compared to the extracellular region of CiTLR19 (docking score = −474.90), the extracellular region of ScTLR19 (docking score = −512.31) exhibited a stronger tendency for molecular docking with polyI:C (Table 1). Furthermore, the extracellular region of ScTLR19 had more potential interaction sites with polyI:C than that of CiTLR19 (Figure 6). Specifically, 22 amino acid residues (ARG-238, SER-285, GLN-314, THR-337, THR-339, LYS-340, ARG-366, GLN-363, GLN-414, LYS-440, LEU-464, ASP-466, ARG-513, ASN-541, ASN-564, GLU-590, ASN-614, ASN-640, HIS-642, ASN-665, HIS-666, and TYR-724) in the extracellular region of ScTLR19 showed potential binding with polyI:C (Figure 6A), whereas only 17 amino acid residues (ARG-238, SER-285, GLN-314, THR-337, THR-339, GLN-363, ARG-366, GLN-414, ARG-513, ARG-515, ASN-564, SER-588, ASN-640, TYR-663, ASN-665, HIS-666, ARG-688, and TYR-742) in the extracellular region of CiTLR19 exhibited potential binding with polyI:C (Figure 6B). In terms of spatial conformation, polyI:C was positioned closer to the extracellular region of ScTLR19 compared to that of CiTLR19 (Figure 6).

To further clarify the impact of the extracellular LRR9 on the polyI:C binding activity of ScTLR19, we replaced the amino acid sequence of the LRR9 (amino acid positions 654–677) in the extracellular region of ScTLR19 with the corresponding sequence from CiTLR19, creating a ScTLR19 LRR9 mutant (Figure 7A). Subsequently, the overexpression of wild-type ScTLR19 and the ScTLR19 LRR9 mutant was conducted by transfection in the *S. curriculus* fin (SCF) cell line, and the binding activity of wild-type ScTLR19 and the ScTLR19 LRR9 mutant to polyI:C was compared using an enzyme-linked immunosorbent assay (ELISA). The ELISA results showed that the negative control group (transfected with an empty vector pEGFP-C1-Flag) exhibited no binding between EGFP and polyI:C (Figure 7B). In contrast, both the wild-type ScTLR19 and ScTLR19 LRR9 mutant demonstrated binding activity to polyI:C (Figure 7B). However, the binding activity of the ScTLR19 LRR9 mutant to polyI:C was significantly weaker than that of the wild-type ScTLR19 (*p* < 0.05, Figure 7B). 

### 2.5. Differences in the Activation of Antiviral Immune Pathways by TLR19s from Barbel Chub and Grass Carp

A previous study has reported that CiTLR19 induces interferon-mediated antiviral immune responses through the TRIF-IRF3 signaling pathway [16]. To investigate whether there are differences in the activation of antiviral immune pathways by ScTLR19 and CiTLR19, the expression levels of the IRF3, IRF7, TRIF, and MyD88 genes were quantitatively analyzed by qPCR during a GCRV challenge after ScTLR19 overexpression in SCF cells (Figure 8). The results showed that after the ScTLR19 overexpression in SCF, the expression levels of IRF3 at 12 and 24 h post-GCRV challenge were significantly higher than those of the control group, reaching 9.6-fold (*p* < 0.01) and 5.3-fold (*p* < 0.01) of the control, respectively (Figure 8A). In contrast, the expression levels of IRF7 at all time points (0, 12, 24, and 48 h) during the GCRV challenge were not significantly different from those of the control group (Figure 8B). After ScTLR19 overexpression, the expression levels of TRIF at 12 and 24 h post-GCRV challenge were significantly higher than those of the control group, with expression levels 3.7-fold (*p* < 0.01) and 2.9-fold (*p* < 0.01) of the control, respectively (Figure 8C). Notably, the expression trend of MyD88 was similar to that of TRIF, with significantly higher expression levels at 12 and 24 h post-GCRV challenge compared to the control group, reaching 3.9-fold (*p* < 0.01) and 4.0-fold (*p* < 0.01) of the control, respectively (Figure 8D). These results indicated that the pathway molecules involved in ScTLR19-mediated interferon antiviral immune responses included IRF3, TRIF, and MyD88.

To further clarify whether the extracellular LRR9 influences the antiviral immune pathway activated by ScTLR19, we compared the regulatory effects of wild-type ScTLR19 and the ScTLR19 LRR9 mutant on the expression of IRF3, TRIF, and MyD88 (Figure 9). Before the GCRV challenge, there were no significant differences in the expression levels of IRF3 (Figure 9A), TRIF (Figure 9B), and MyD88 (Figure 9C) between the ScTLR19 LRR9 mutant group and the wild-type ScTLR19 group (*p* > 0.05). However, 24 h after the GCRV challenge, the expression levels of IRF3 (Figure 9A), TRIF (Figure 9B), and MyD88 (Figure 9C) in the ScTLR19 LRR9 mutant group were significantly lower than those in the wild-type ScTLR19 group (*p* < 0.01).

## 3. Discussion

During long-term evolution, the same PRR gene in different fish species has undergone structural and functional differentiation, leading to the induction of distinct immune responses when they recognize the same viral infection [30,32]. This may be one of the reasons for the differences in the antiviral capabilities among different fish species. Therefore, comparing the structural and functional differences in the PRR gene in different fish species is of great significance for revealing the mechanisms underlying their disease resistance. Grass carp (*C. idella*), the most widely cultured freshwater fish globally, is highly susceptible to GCRV infection, suffering from hemorrhagic disease [33]. Barbel chub (*S. curriculus*), which belongs to the same subfamily Leuciscinae as grass carp, exhibits strong resistance to GCRV infection [27]. Moreover, the hybrid offspring of barbel chub and grass carp show stronger resistance than grass carp, making barbel chub the ideal model for studying GCRV resistance mechanisms and the genetic basis of resistance in barbel chub and grass carp hybrid offspring [34]. TLRs play indispensable roles in antiviral immunity, while TLR19 is a fish-specific TLR [15,16]. To date, studies on fish TLR19s are not systematic and in-depth, and most of them are focused on molecular identification and expression level analysis [18]. This study, for the first time, discovered significant differences in the structures and immune functions of TLR19s from barbel chub and grass carp, providing clues to the molecular mechanisms underlying the differences in GCRV resistance in these two fish.

The results of this study show that both ScTLR19 and CiTLR19 encode 975 amino acids and possess an extracellular LRR region, a transmembrane domain, and an intracellular TIR domain. Although sequence and phylogenetic tree analyses indicate high sequence similarity and homology between ScTLR19 and CiTLR19, the domain prediction results reveal that ScTLR19 has one additional LRR (LRR9) at amino acid positions 654–677 in the extracellular region compared to CiTLR19. Correspondingly, sequence alignment and three-dimensional structure comparisons show that the extracellular region of ScTLR19 exhibits greater structural differences from CiTLR19 than the intracellular region. The extracellular LRR region, a pathogen recognition motif present in all TLRs, consists of α-helices and β-sheets that form loops and connections [35]. During TLR-mediated pathogen recognition, LRRs form a horseshoe-shaped non-globular structure, with a series of LRRs creating a β-sheet concave surface that serves as a site for molecular interactions, closely related to TLR ligand binding [36,37]. To some extent, the structural features of the extracellular LRR region determine the ligand-binding activity, i.e., the immune recognition function of TLRs [37,38]. This study suggests that the differences in the extracellular LRR structures of ScTLR19 and CiTLR19 are likely responsible for their differences in immune recognition functions.

PolyI:C, a double-stranded RNA viral analog, is a common ligand used to study the viral recognition function of TLRs [39]. In mammals, studies have shown that TLR3’s extracellular region contains 23 LRRs (LRR1–LRR23), with LRR20 identified as the critical motif for TLR3 binding to polyI:C [40]. In fish, molecular modeling has predicted that rohu TLR3 has 27 LRRs, with LRR4–6, 13–14, and 20–22 being key motifs for polyI:C binding, while LRR17–24 may be crucial for binding dsRNA from fish reoviruses [41]. Additionally, two SNP sites (p.L159F and p.L529P) in the LRR region have been found to affect the polyI:C binding activity of rohu TLR22 and alter the antiviral capacity of fish [42]. In this study, molecular docking analysis revealed that the binding sites of ScTLR19’s extracellular region to polyI:C differ significantly from those of CiTLR19. Overall, ScTLR19’s extracellular region has more binding sites for polyI:C. Furthermore, the potential affinity of ScTLR19’s extracellular region for polyI:C (docking score = −512.31) is stronger than that of CiTLR19’s extracellular region (docking score = −474.90). In terms of spatial conformation, polyI:C is positioned closer to the extracellular region of ScTLR19 compared to CiTLR19. More importantly, our mutant experiments demonstrated that replacing LRR9 in ScTLR19 with the corresponding amino acid sequence from CiTLR19 reduced its polyI:C binding activity, confirming that LRR9 is a critical motif for ScTLR19’s binding to polyI:C. Although this study did not directly compare the polyI:C binding activity of ScTLR19 and CiTLR19, the molecular docking and mutant experiment results collectively suggest that ScTLR19 may have stronger polyI:C binding activity than CiTLR19.

IRF3 and IRF7, members of the IRF family, are both inducers of type I interferon (IFN) and play key roles in TLR-mediated antiviral immunity [43,44]. As transcription factors, IRF3 and IRF7 remain inactive in the cytoplasm of host cells in the absence of viral stimulation [45,46]. Upon viral invasion, IRF3 and IRF7 are activated and phosphorylated by upstream molecules in the TLR signaling pathway and then translocated into the nucleus to regulate interferon expression and combat the invading viruses [47,48,49]. A previous study has shown that after GCRV infection, CiTLR19 only regulates IRF3, but not IRF7 [16]. In this study, after the overexpression of ScTLR19 in SCF cells and GCRV infection, the expression level of IRF3 was significantly upregulated compared to that of the control group, while IRF7 expression showed no significant change. This indicates that in the anti-GCRV immune response of barbel chub, ScTLR19 also only regulates IRF3 but not IRF7. Therefore, the regulatory and activation patterns of the IRF family by ScTLR19 and CiTLR19 in the anti-GCRV immune response may be consistent. 

TLR-mediated signaling cascades are primarily divided into MyD88-dependent and TRIF-dependent pathways [50,51]. MyD88, the first identified adaptor protein mediating TLR signaling, recruits interleukin-1 receptor-associated kinases and ultimately activates downstream factors in the TLR signaling pathway [52,53]. Studies have shown that the overexpression of MyD88 in black carp *Mylopharyngodon piceus* kidney cells followed by *Aeromonas hydrophila* stimulation activates the NF-κB signaling pathway and significantly upregulates the expression of tumor necrosis factor-α (TNF-α), IFN-α, and interleukin-6 (IL-6) [54]. The knockdown of MyD88 significantly reduces the expression levels of these factors [54]. Additionally, the overexpression of lamprey *Lethenteron camtschaticum* MyD88 in HEK293T cells enhances NF-κB activity, initiating the TLR signaling pathway [55]. Therefore, MyD88 has a conserved function in fish TLR signaling pathways. In MyD88-independent TLR-mediated signaling cascades, the adaptor protein TRIF can replace MyD88 to play a critical role. TRIF, also known as toll-like receptor signaling adaptor molecule 1 (TICAM-1), first binds to TNF receptor-associated factors (TRAFs) at its N-terminus, and then it interacts with the kinase TBK1 (TANK-binding kinase-1), ultimately inducing IFN expression [56]. Mammalian TLR3- and TLR4-mediated immune signaling pathways are mediated by TRIF, which activates the IRF pathway and induces IFN expression to transduce upstream immune signals [57,58]. In this study, after the overexpression of ScTLR19, the GCRV challenge significantly upregulated the expression levels of both MyD88 and TRIF, suggesting that in the ScTLR19 signaling pathway, ScTLR19 may regulate both MyD88 and TRIF to synergistically exert immune functions. In contrast, it has been reported that CiTLR19 only activates TRIF to induce downstream immune signals [16]. From this perspective, the anti-GCRV immune signals mediated by ScTLR19 may be more extensive than by CiTLR19.

Previous studies have reported that the extracellular LRR region not only determines the pathogen recognition function of TLRs but also influences their ability to activate immune pathways [40]. In humans, the deletion of LRR20, 21, or 22 in TLR3 reduces the oligomerization of the intracellular TIR domain, thereby affecting TLR3 pathway activation and interferon-mediated antiviral immune responses [40]. In fish, structural variations in the extracellular LRR region have also been found to affect the interaction between TLR3 and TRIF [41]. In this study, given that ScTLR19 has one additional LRR9 in the extracellular region compared to CiTLR19, we hypothesized that this structural difference might lead to differences in their anti-GCRV immune signaling pathways. Subsequent mutant experiments showed that replacing LRR9 in ScTLR19 with the corresponding amino acid sequence from CiTLR19 altered the expression regulation on IRF3, MyD88, and TRIF after GCRV infection. Therefore, LRR9 may be the key molecular basis for both the stronger viral recognition ability of ScTLR19 compared to CiTLR19 and the broader antiviral immune signals mediated by ScTLR19.

In conclusion, this study identified a TLR19 homolog in barbel chub *S. curriculus* and compared its structural and functional differences with the previously reported CiTLR19 in grass carp *C. idella* (Figure 10). Structurally, ScTLR19 has one additional LRR9 in the extracellular region compared to CiTLR19 (Figure 10A). Functionally, the presence of LRR9 enhances the polyI:C binding activity of ScTLR19 and enables it to mediate antiviral immune signaling pathways that are different from those of CiTLR19. ScTLR19 can simultaneously regulate the IRF3-TRIF and IRF3-MyD88 signaling pathways (Figure 10B), which may allow ScTLR19 to mediate more extensive anti-GCRV immune signals and functional roles, thereby enhancing the GCRV resistance of barbel chub. These findings may provide the key basis for the further exploration of the antiviral functions of fish TLR19s and the mechanisms underlying the different strategies employed by barbel chub and grass carp in response to the GCRV and the genetic basis of GCRV resistance in the hybrid offspring of barbel chub and grass carp.

## 4. Materials and Methods

### 4.1. Cell Line and Viruses

The *S. curriculus* fin (SCF) cell line previously established by our laboratory [59] was cultured in M199 medium (Hyclone, Logan, UT, USA) supplemented with 10% fetal bovine serum (Gibco, Carlsbad, CA, USA) and 1% penicillin-streptomycin at 28 °C with a CO_2_ concentration of 5%.

GCRV-I (strain JX-0901) can induce large syncytia in different fish cell lines, which was kindly gifted by Prof. Cunbin Shi from the Pearl River Fisheries Research Institute of the Chinese Academy of Fishery Sciences (Guangzhou, China). The GCRV JX-0901 strain (1.0 × 10^3.6^ TCID_50_/mL) was applied for the cell challenge experiment in vitro.

### 4.2. Gene Cloning of ScTLR19 by RACE

Using the sequence information of the CiTLR19 gene (GenBank: KY798320.1) from NCBI (https://www.ncbi.nlm.nih.gov; accessed on 6 March 2022) as a reference, primers ScTLR19-cF/cR (Table 2) were designed using Oligo 7.0 software. The middle fragment of the ScTLR19 gene was amplified by PCR using cDNA from the liver of barbel chub as a template. The PCR reaction system consisted of 10 μL Taq Master Mix (TaKaRa, Dalian, China), 7 μL ddH_2_O, 1 μL cDNA, 1 μL ScTLR19-cF, and 1 μL ScTLR19-cR. The PCR program was as follows: 94 °C for 5 min; 35 cycles of 94 °C for 30 s, 56 °C for 30 s, and 72 °C for 2 min; followed by 72 °C for 7 min; and storage at 4 °C. The PCR product was ligated into the pMD19-T vector (TaKaRa) and then sent to Sangon Biotech (Shanghai, China) for sequencing.

For the 3′ RACE first round, ScTLR19 3′ W and the universal primer UPM (Table 2) were used as amplification primers. The PCR conditions were as follows: 94 °C for 5 min; 35 cycles of 94 °C for 30 s, 55 °C for 30 s, and 72 °C for 2 min; followed by 72 °C for 7 min. The first-round PCR product was then diluted 50-fold and used as a template for the nested PCR using ScTLR19 3′ N and the universal primer UPM (Table 2). For the 5′ RACE first round, ScTLR19-5′ and the universal primer UPM (Table 2) were used as amplification primers. The amplification conditions were as follows: 94 °C for 5 min; 35 cycles of 94 °C for 30 s, 60 °C for 30 s, and 72 °C for 2 min; followed by 72 °C for 7 min. The first-round PCR product was then diluted 50-fold and used as a template for the nested PCR. By comprehensively analyzing the sequences obtained from 3′ RACE, 5′ RACE, and the middle fragment, the full-length cDNA sequence of the ScTLR19 gene was obtained.

### 4.3. Sequence Analysis of TLR19s

The deduced amino acid sequence of ScTLR19 was analyzed using the Expert Protein Analysis System (https://web.expasy.org/translate/; accessed on 14 March 2022). The isoelectric points and molecular weights of ScTLR19 were predicted using the ExPASy platform (https://web.expasy.org/compute_pi/; accessed on 14 March 2022). The signal peptides of TLR19s were predicted using SignalP 6.0 (https://services.healthtech.dtu.dk/services/SignalP-6.0/; accessed on 14 March 2022), and transmembrane regions were predicted using TMHMM Server v.2.0 (https://www.hsls.pitt.edu/obrc/index.php?page=URL1164644151; accessed on 14 March 2022). Sequence similarity and homology among 9 fish TLR19s and mouse TLR11 were calculated using MATGAT 2.0 software [60]. The protein domains of 6 fish TLR19s were identified using the Simple Modular Architecture Research Tool (http://smart.emblheidelberg.de/; accessed on 14 March 2022), and their multiple sequence alignment was performed using DNA Man 6.0 software (Lynnon Biosoft Inc., Burlington, NC, USA). All the accession information of TLRs used for sequence analysis is listed in Table 3.

### 4.4. Phylogenetic Analysis of TLR19s

A phylogenetic tree was constructed by using MEGA software Version 7.0 (Pennsylvania State University, State College, PA, USA) based on the full-length amino acid sequences of 9 fish TLR19s and all TLR1-11s from humans or mice, and 1000 bootstrap replicates were set to assess the reliability of the branching. The phylogenetic tree was displayed and annotated by using iTOL (https://itol.embl.de/; accessed on 21 May 2023). The information of TLRs from different vertebrates used in the phylogenetic analysis is listed in Table 3.

### 4.5. Molecular Modeling and Docking

The three-dimensional structures of CiTLR19 and ScTLR19, and their extracellular domain (ECD) and intracellular domain (ICD), were generated using the Swiss-Model server (https://swissmodel.expasy.org/; accessed on 18 July 2024) with A0A2U7PQL2 and A0A6M6A8Q3 in the AlphaFold Protein Structure Database (https://alphafold.com/; accessed on 18 July 2024) as templates, respectively. All the obtained models were optimized by Modeller (https://salilab.org/modeller/; accessed on 18 July 2024) and assessed by SAVES v6.1 (https://saves.mbi.ucla.edu/; accessed on 18 July 2024), ProQ (https://proq.bioinfo.se/cgi-bin/ProQ/ProQ.cgi; accessed on 18 July 2024), ProSA (https://prosa.services.came.sbg.ac.at/prosa.php), MolProbity (http://molprobity.biochem.duke.edu/; accessed on 18 July 2024), and VADAR V1.8 (http://vadar.wishartlab.com/index.html; accessed on 18th July 2024). The three-dimensional structures of polyI:C were generated and energy-minimized using ChemDraw 23.1 and Chem3D software v14.0 (https://www.chemdraw.com.cn/; accessed on 23 July 2024) based on a 2-D structure (SID: 53787079) in the PubChem database (https://pubchem.ncbi.nlm.nih.gov/; accessed on 23th July 2024). Molecular docking was performed using the HDOCK server (http://hdock.phys.hust.edu.cn/; accessed on 23 July 2024) and evaluated using the PLIP server (https://plip-tool.biotec.tu-dresden.de/plip-web/plip/index; accessed on 23 July 2024). The visualizations of protein structures and molecular docking were conducted with PyMol software 3.0 (https://www.pymol.org/; accessed on 27 July 2024). 

### 4.6. The Construction of ScTLR19 and ScTLR19 LRR9 Mutant Vectors

Using the ScTLR19 ORF sequence as a template, the specific primers ScTLR19-oF/oR containing restriction enzyme sites (5′ Kpn I and 3′ EcoR I) were designed to conduct a PCR. Then, the PCR product was ligated into the pEGFP-N1-Flag vector to construct the recombinant plasmid pEGFP-N1-Flag-ScTLR19. The ORF encoding the ScTLR19 LRR9 mutant was constructed by Sangon Biotech (Shanghai, China) with gene synthesis, and the DNA sequence encoding the LRR9 (amino acid positions 654–677) in the extracellular region of ScTLR19 was replaced with the corresponding sequence from CiTLR19. Then, the ORF encoding the ScTLR19 LRR9 mutant was ligated into the pEGFP-N1-Flag vector to construct the recombinant plasmid pEGFP-N1-Flag-ScTLR19 LRR9 mutant. The recombinant pEGFP-N1-Flag-ScTLR19 plasmids or the pEGFP-N1-Flag-ScTLR19 LRR9 mutant were extracted according to the instructions of the Endo-Free Plasmid Mini Kit I (TaKaRa). The concentration of the plasmids was measured by Nanodrop 2000 (ThermoFisher Scientific, Waltham, MA, USA) and stored at −20 °C for use.

### 4.7. Transfection

The overexpression of ScTLR19 or the ScTLR19 LRR9 mutant was carried out in SCF cells by transfection. Transfection was executed using the Lipofectamine 2000 reagent (Invitrogen, Karlsbad, CA, USA) according to the manufacturer’s instructions. Briefly, SCF cells were seeded into 6-well culture plates containing M199 medium (Hyclone) supplemented with 10% fetal bovine serum (Gibco) and 1% penicillin-streptomycin at 28 °C with a CO_2_ concentration of 5%. When the cells covered 70–90% of the well, 1 or 2 μg of either pEGFP-N1-Flag-ScTLR19, the pEGFP-N1-Flag-ScTLR19 LRR9 mutant, or pEGFP-C1-Flag (negative control) plasmids and FuGENE HD transfection reagent (Promega, Madison, WI, USA) were added. After 48 h, SCK cells overexpressing ScTLR19 or the ScTLR19 LRR9 mutant were examined under a fluorescence microscope (Olympus BX51, Tokyo, Japan). The SCF cells overexpressing ScTLR19 or the ScTLR19 LRR9 mutant were used in the subsequent ELISA for analyzing the binding activity of TLR19s to polyI:C and the in vitro GCRV challenge experiment. 

### 4.8. ELISA

An ELISA was performed to analyze differences in the binding activity of ScTLR19 and the ScTLR19 LRR9 mutant to polyI:C. First, the 96-well microtiter plates (Costar, Corning, Armonk, NY, USA) were coated with polyI:C (10 μg/well, Sigma-Aldrich, St. Louis, MO, USA) in 100 μL of carbonate–bicarbonate buffer (50 mM, pH 9.6) at 4 °C overnight. After three times of washing with PBST (137 mM NaCl, 2.7 mM KCl, 10 mM Na_2_HPO_4_, 2 mM KH_2_PO_4_, and 0.05% Tween-20, pH 7.4), the wells were blocked with 200 μL of 5% bovine serum albumin solution. The SCF cells overexpressing ScTLR19, the ScTLR19 LRR9 mutant, or EGFP (negative control) with the Flag tag were lysed by the RIPA Lysis Buffer (Beyotime Biotechnology, Shanghai, China) on ice, followed by centrifugation at 8000× *g*, 4 °C for 10 min to obtain the supernatant. The total protein concentration of the supernatant was determined according to a BCA assay kit (Sigma-Aldrich) and adjusted into 5 mg/mL with the RIPA Lysis Buffer. The above supernatant was added into the 96-well microtiter plates with 100 μL per well for 3 h. The rabbit anti-Flag tag polyclonal antibody (ABclonal, diluted 1:1000) and the second antibody (HRP-labeled goat anti-rabbit IgG, Abcam, diluted 1:3000) were added into the wells and incubated for 1 h. After extensive washing, 100 μL of Soluble TMB Substrate Solution (Tiangen, Beijing, China) was added and incubated at 37 °C in the dark for 30 min. The reaction was stopped by adding 50 μL of 2 M H_2_SO_4_ solution per well, and the absorbance value was measured using a microplate reader at 450 nm (Tecan, San Jose, CA, USA). Wells with 100 μL of TBS were used as a blank. Each group was set up with 6 replicates.

### 4.9. In Vitro GCRV Challenge Experiment

The in vitro GCRV challenge experiment was carried out in SCF cells to investigate the activation of antiviral immune signal pathway by ScTLR19 or the ScTLR19 LRR9 mutant. The SCF cells overexpressing ScTLR19 or the ScTLR19 LRR9 mutant were cultured in 6-well culture plates at 28 °C and were challenged with the GCRV for 48 h. The cells were collected at 0, 12, 24, and 48 h after the GCRV challenge. The SCF cells transfected with the empty vector pEGFP-C1-Flag and challenged with the GCRV were set as the control. Three repeated cell samples were taken at each time point.

### 4.10. RNA Extraction, cDNA Synthesis, and Quantitative PCR (qPCR)

The total RNA of the above cell samples was extracted using TRIzol reagent (ThermoFisher Scientific). First-strand cDNA synthesis was performed using the RevertAid First Strand cDNA Synthesis Kit (ThermoFisher Scientific) according to the manufacturer’s instructions. qPCR was performed to evaluate the mRNA expression levels of the target genes, with the specific primers listed in Table 2. The qPCR analysis was conducted using the CFX96 Touch Real-Time PCR Detection System (Bio-Rad, Hercules, CA, USA) with a total reaction volume of 10 μL, containing 5 μL of SYBR qPCR Master Mix (Vazyme, Nanjing, China), 2 μL of diluted cDNA template (1:10), 0.8 μL of primer mix, and 2.2 μL of ddH_2_O. β-actin was used as the reference gene, and the relative mRNA expression levels of the target genes were calculated using the 2^−ΔΔCt^ method [61].

### 4.11. Statistical Analysis

Data were presented as mean ± standard deviation unless otherwise indicated. The two-sample Student’s *t*-test was performed for the comparisons between groups using the Statistical Package for Social Sciences Version 25.0 (SPSS Inc., Chicago, IL, USA). *p* < 0.05 was considered statistically significant.

## Figures and Tables

**Figure 1 ijms-26-03103-f001:**
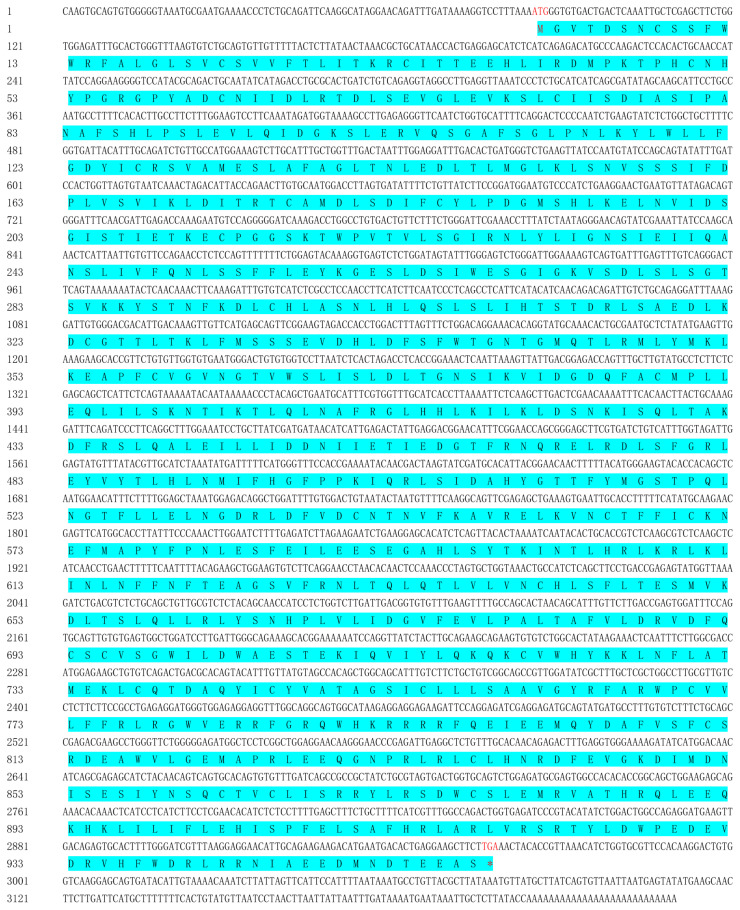
The cDNA and amino acid sequences of ScTLR19. The start codon is highlighted in red, while the stop codon is marked with a red asterisk (*). The encoding amino acid sequence is indicated with light blue shading.

**Figure 2 ijms-26-03103-f002:**
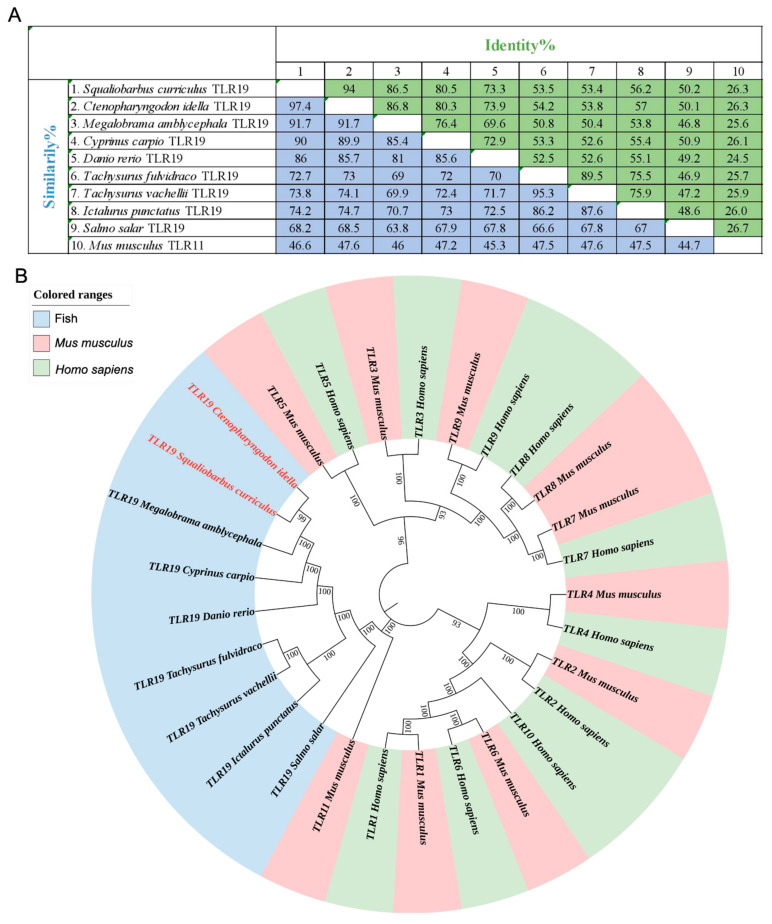
The amino acid sequence identity and similarity and phylogenetic tree of TLRs from various vertebrates. (**A**) The amino acid sequence identity and similarity for 9 fish TLR19s and mouse TLR11. The values of similarity and identity are backgrounded in blue and green, respectively. (**B**) The phylogenetic tree was constructed using the neighbor-joining method and MEGA 6 software based on TLRs including 9 fish TLR19s’ protein sequences from different vertebrate species. The ScTLR19 and CiTLR19 are indicated by the red font. TLRs from fish, humans, and mice are marked by different colors, and their accession information is shown in the Materials and Methods.

**Figure 3 ijms-26-03103-f003:**
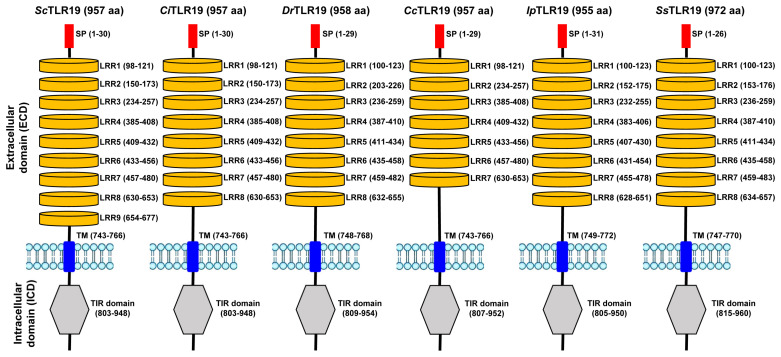
Protein domain composition of six fish TLR19s. The selected TLR19s were from barbel chub (*Sc*), grass carp (*Ci*), common carp (*Cc*), channel catfish (*Ip*), zebrafish (*Dr*), and Atlantic salmon (*Ss*), and their accession information is shown in the Materials and Methods. SP: signal peptide; TM: transmembrane domain.

**Figure 4 ijms-26-03103-f004:**
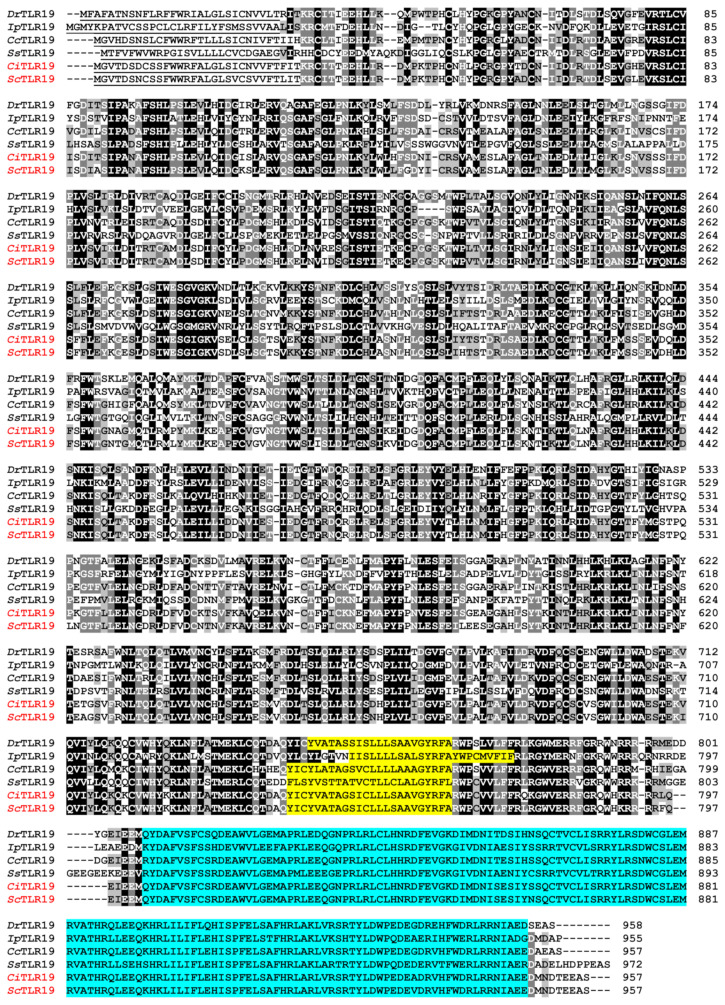
**The** multiple sequence alignment of CiTLR19 and ScTLR19 with vertebrates’ counterparts. The accession information of six selected fish TLR19s is shown in the Materials and Methods. The amino acid sequences encoding SPs are bold and underlined. The amino acid sequences encoding the TM and TIR domains are highlighted in yellow and turquoise, respectively.

**Figure 5 ijms-26-03103-f005:**
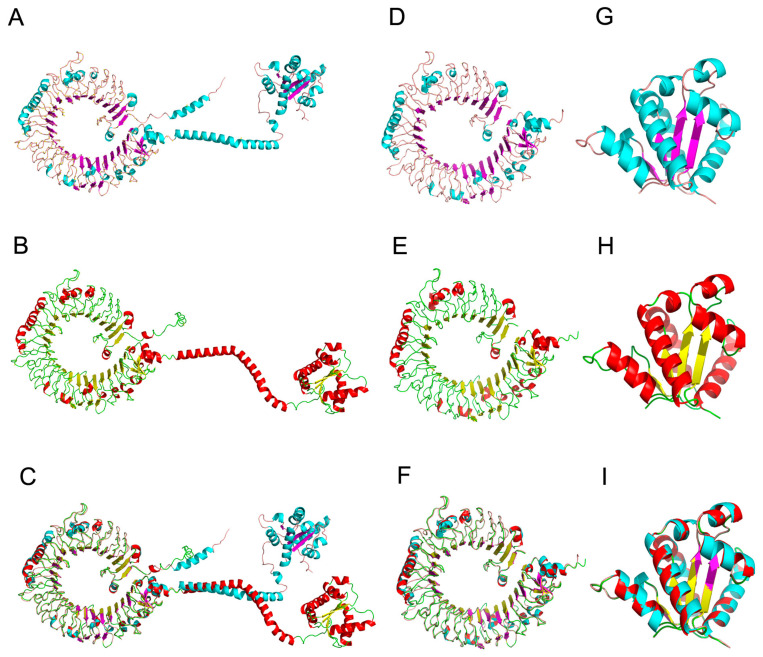
The three-dimensional structures of ScTLR19 and CiTLR19. The three-dimensional structures of (**A**) ScTLR19 and (**B**) CiTLR19 generated using a Swiss-Model server with A0A2U7PQL2 and A0A6M6A8Q3 in the AlphaFold Protein Structure Database as templates. (**C**) The structural alignment of the ScTLR19 and CiTLR19 models using PyMol software. The three-dimensional structures of (**D**) the ScTLR19 extracellular domain (ECD) and (**E**) thCe iTLR19 ECD generated using the Swiss-Model server with A0A2U7PQL2 and A0A6M6A8Q3 in the AlphaFold Protein Structure Database as templates. (**F**) The structural alignment of the ScTLR19 ECD and CiTLR19 ECD models using PyMol software. The three-dimensional structures of (**G**) the ScTLR19 intracellular domain (ICD) and (**H**) the CiTLR19 ICD generated using the Swiss-Model server with A0A2U7PQL2 and A0A6M6A8Q3 in the AlphaFold Protein Structure Database as templates. (**I**) The structural alignment of the ScTLR19 ICD and CiTLR19 ICD models using PyMol software.

**Figure 6 ijms-26-03103-f006:**
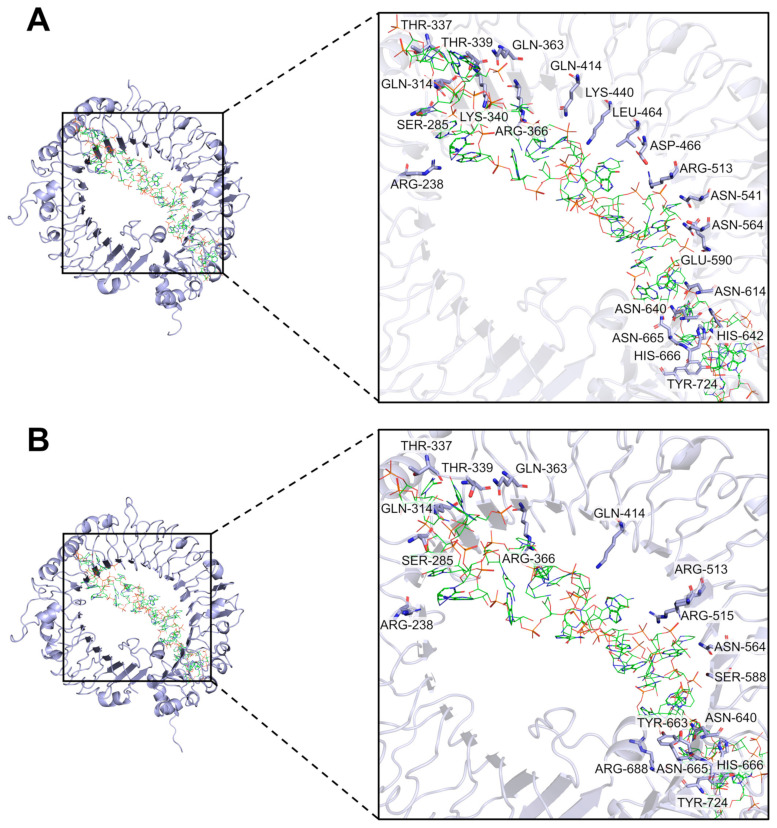
The molecular docking of ScTLR19-ECD (**A**) and CiTLR19-ECD (**B**) with polyI:C. The three-dimensional structures of ScTLR19 ECD, CiTLR19 ECD, and polyI:C were generated using the Swiss-Model server, ChemDraw 23.1, and Chem3D software v14.0. After the quality of the models was verified, the molecular docking was conducted using the HDOCK server. Red and green sticks stand for polyI:C. The α-helices, β-sheets, and loops of ScTLR19-ECD and CiTLR19-ECD are shown with the blue color, and the potential interaction sites of ScTLR19-ECD and CiTLR19-ECD to polyI:C are marked.

**Figure 7 ijms-26-03103-f007:**
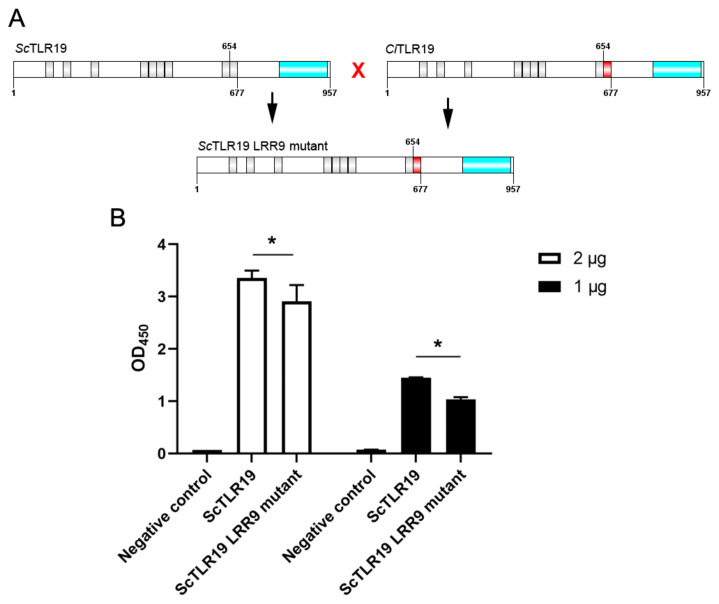
Differences in the binding activity of ScTLR19 and the ScTLR19 LRR9 mutant to polyI:C, analyzed by an ELISA. (**A**) The schematic diagram for the construction of the ScTLR19 LRR9 mutant. The ORF encoding the ScTLR19 LRR9 mutant was constructed by Sangon Biotech with gene synthesis, and the DNA sequence encoding the LRR9 (amino acid positions 654–677) in the extracellular region of ScTLR19 was replaced with the corresponding sequence from CiTLR19. (**B**) The ELISA for detecting the differences in the binding activity of ScTLR19 and the ScTLR19 LRR9 mutant to polyI:C. The OD_450_ values reflect the binding activity of ScTLR19,the ScTLR19 LRR9 mutant, and EGFP (negative control) to polyI:C. The ScTLR19, ScTLR19 LRR9 mutant, and EGFP proteins were expressed by transfection with pEGFP-N1-Flag-ScTLR19, the pEGFP-N1-Flag-ScTLR19 LRR9 mutant, or pEGFP-C1-Flag plasmids (1 or 2 μg per well in 6-well culture plates) in SCF cells, respectively. “*” represents statistical significance at *p* < 0.05 (N = 6).

**Figure 8 ijms-26-03103-f008:**
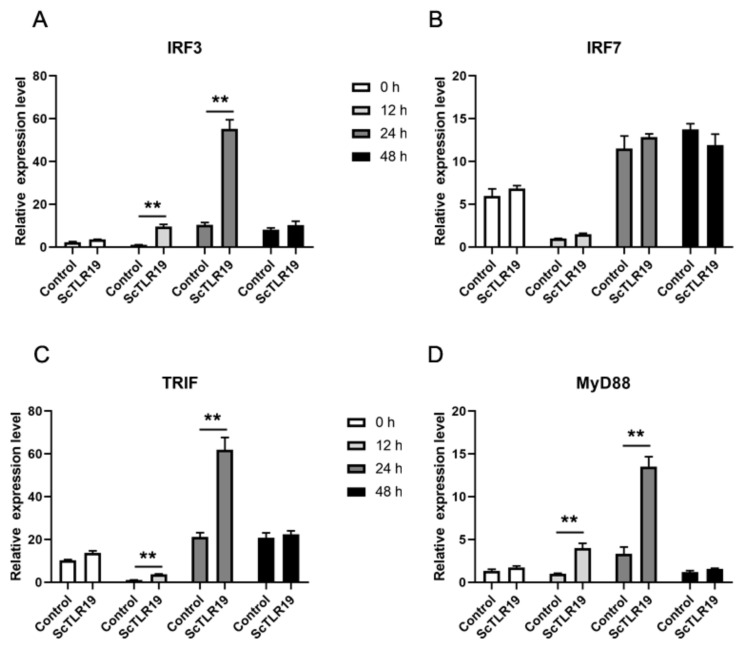
The activation of antiviral immune pathway genes by ScTLR19. The mRNA expression levels of the (**A**) IRF3, (**B**) IRF7, (**C**) TRIF, and (**D**) MyD88 genes were quantitatively analyzed by qPCR after a GCRV challenge at 0, 12, 24, 48 h in the SCF cells overexpressing ScTLR19. The SCF cells transfected with the empty vector pEGFP-C1-Flag and challenged with the GCRV were set as the control. “**” represents statistical significance at *p* < 0.01 (N = 3).

**Figure 9 ijms-26-03103-f009:**
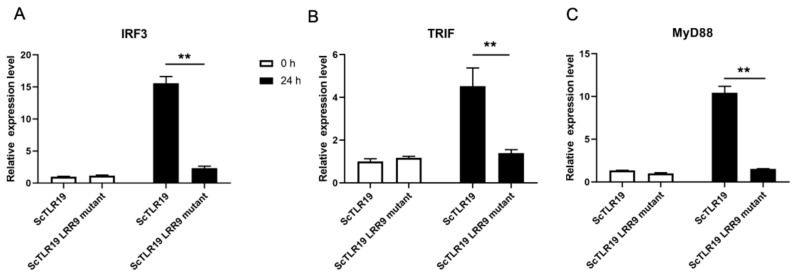
Differences in the activation of antiviral immune pathway genes by ScTLR19 and the ScTLR19 LRR9 mutant. The mRNA expression levels of the (**A**) IRF3, (**B**) TRIF, and (**C**) MyD88 genes were quantitatively analyzed by qPCR after the GCRV challenge at 0, 24 h in the SCF cells overexpressing ScTLR19 or the ScTLR19 LRR9 mutant. “**” represents statistical significance at *p* < 0.01 (N = 3).

**Figure 10 ijms-26-03103-f010:**
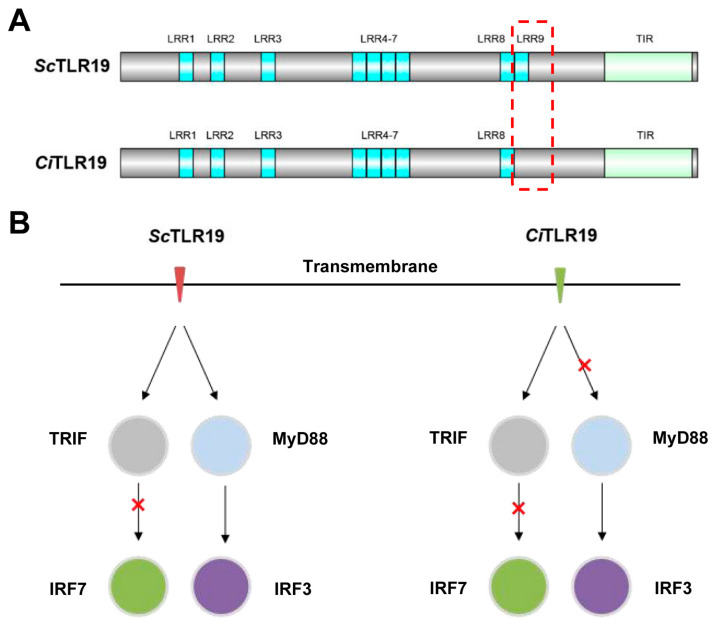
The schematic diagram for the differences in the structural and antiviral immune pathways of ScTLR19 and CiTLR19. (**A**) ScTLR19 has one additional LRR9 in the extracellular region compared to CiTLR19. (**B**) ScTLR19 regulates both the IRF3-TRIF and IRF3-MyD88 signaling pathways during GCRV infection, while CiTLR19 only regulated the IRF3-MyD88 signaling pathway.

**Table 1 ijms-26-03103-t001:** Summary of molecular docking results.

	CiTLR19-ECD and Poly (I:C)	ScTLR19-ECD and Poly (I:C)
Docking score	−474.90	−512.31
Confidence score	0.9985	0.9993

Notes: a more negative docking score means a more possible binding model; confidence score > 0.7 (the two molecules would be very likely to bind), 0.5 < confidence score < 0.7 (the two molecules would be possible to bind), confidence score < 0.5 (the two molecules would be unlikely to bind).

**Table 2 ijms-26-03103-t002:** Primers and their sequences used in this study.

Primers	Sequences (5′-3′)	Applications
ScTLR19 5′	TCCAGGTGGTCTACTTCCGAACTGCTC	RACE
ScTLR19 3′W	ACTAATGTTTTCAAGGCAGTTCGAGAGC	RACE
ScTLR19 3′N	CACTGCACCGTCTCAAGCGTCT	RACE
UPM	CTAATACGACTCACTATAGGGC	RACE
ScTLR19-cF	CTGCCAATGCCTTTTCACAC	PCR
ScTLR19-cR	GCAAACAGAGCCTCAATCTCG	PCR
ScTLR19-oF	tcgagctcaagcttcgaattcATGGGTGTGACTGACTCAAATTGC	Construction of overexpression vector
ScTLR19-oR	ggatcccgggcccgcggtaccAGAAGCTTCCTCAGTGTCATTCATG	Construction of overexpression vector
ScTLR19-qF	TTCACACTTGCCTTCTTTGGA	qPCR
ScTLR19-qR	TAATCACCGAAAAGCAGCCAGA	qPCR
MyD88-qF	GAAGCCCGTCCAGGTTCCA	qPCR
MyD88-qR	TGAAGGCATCAAAGGTCTCCG	qPCR
TRIF-qF	TCGAGGAGGAAGGCTGGCACTT	qPCR
TRIF-qF	TGAGCGTGAAGCAGACAGCAGC	qPCR
IRF7-qF	CGCCTGTGTTCGTCACTCGT	qPCR
IRF7-qR	GGTGGTTGGAAAGCGTATTGG	qPCR
IRF3-qF	AAACGTGTTCTGTACAAACCC	qPCR
IRF3-qR	CTCAAGCCCCAAATAGAGCTG	qPCR
β-actin-qF	GCTATGTGGCTCTTGACTTCG	qPCR
β-actin-qR	GGGCACCTGAACCTCTCATT	qPCR

**Table 3 ijms-26-03103-t003:** The information of TLRs from different vertebrates used in sequence and phylogenetic analysis.

Species	TLRs	Accession Numbers
*Tachysurus fulvidraco*	TLR19	QLL99515.1
*Ctenopharyngodon idella*	TLR19	AUF71965.1
*Squaliobarbus curriculus*	TLR19	QJX15323.1
*Ictalurus* *punctatus*	TLR19	AEI59675.1
*Danio rerio*	TLR19	NP001352353.1
*Tachysurus vachellii*	TLR19	WJQ78152.1
*Salmo* *salar*	TLR19	CDH93609.2
*Cyprinus carpio*	TLR19	BAU98390.1
*Megalobrama amblycephala*	TLR19	APT35508.1
*Homo sapiens*	TLR1	CAG38593.1
*Homo sapiens*	TLR2	AAH33756.1
*Homo sapiens*	TLR3	ABC86910.1
*Homo sapiens*	TLR4	AAF05316.1
*Homo sapiens*	TLR5	AAI09119.1
*Homo sapiens*	TLR6	BAA78631.1
*Homo sapiens*	TLR7	AAZ99026.1
*Homo sapiens*	TLR8	AAZ95441.1
*Homo sapiens*	TLR9	AAZ95520.1
*Homo sapiens*	TLR10	AAY78491.1
*Mus musculus*	TLR1	NP001263374.1
*Mus musculus*	TLR2	NP036035.3
*Mus musculus*	TLR3	AAH99937.1
*Mus musculus*	TLR4	NP067272.1
*Mus musculus*	TLR5	AAI25248.1
*Mus musculus*	TLR6	BAA78632.1
*Mus musculus*	TLR7	AAI32386.1
*Mus musculus*	TLR8	AAI32055.1
*Mus musculus*	TLR9	EDL21125.1
*Mus musculus*	TLR11	NP991388.2

## Data Availability

The data presented in this study are available on request from the corresponding author.

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
