# Peer review of "Structural and Functional Characteristics of TLR19 in Barbel Chub Compared to TLR19 in Grass Carp"

_ijms, 2025, doi:10.3390/ijms26073103_

Round 1
Reviewer 1 Report
Comments and Suggestions for Authors
Toll-like receptors (TLRs) play indispensable roles in antiviral immunity of fish. In this study, the structures and antiviral immune functions of barbel chub TLR19 (ScTLR19) and grass carp TLR19 (CiTLR19) were compared by Lv et al. They found that ScTLR19 has one additional LRR9 in the extracellular region compared to CiTLR19, and the presence of LRR9 enhances the polyI:C binding activity and the activation of antiviral immune pathways of ScTLR19. These findings provide a new insight into the mechanisms underlying the different strategies employed by barbel chub and grass carp in response to viral infection and the genetic basis of disease resistance in the hybrid offspring of barbel chub and grass carp. The study is innovative, and the manuscript is well written. I recommend a minor revision, and several issues should be addressed before acceptance.
--Abstract
- The importance of TLRs in antiviral immunity of fish needs to be introduced in Abstract.
- The first letter “Docking Score” should be lowercase.
--Induction
- Line 75: What does “their” stand for?
- The barbel chub anf the hybrid offspring of barbel chub and grass carp exhibit stronger resistance to grass carp reovirus (GCRV) infection than grass carp. Is there any reported data to support it before?
--Results
- Line 108, the word “the” before ScTLR19 should be remove.
- Lines 119-120, “similarity of” should be replaced with “amino acid similarity of”.
- Lines 129 and 131, TLR19 should be replaced with TLR19s.
- Line 152, complete should be deleted.
- Line 157-158, please note the singular and plural forms.
- Lines 212, 213, 349 and 350, the first letter “Docking” should be lowercase.
- Lines 269 and 364, “studies” should be replaced with “study”.
--Discussion
- Lines 343-344, the authors should provide the full names of AGCRV, VHSV, and IHNV.
- Line 359, the authors should provide the full name of type I IFN.
- Line 410, homologs should be replaced with homolog.
--Materials and Methods
- There should be a space between numbers and °C.
- Line 534, 4C?
Comments on the Quality of English Language
Good.
Author Response
Reviewer #1:
Toll-like receptors (TLRs) play indispensable roles in antiviral immunity of fish. In this study, the structures and antiviral immune functions of barbel chub TLR19 (ScTLR19) and grass carp TLR19 (CiTLR19) were compared by Lv et al. They found that ScTLR19 has one additional LRR9 in the extracellular region compared to CiTLR19, and the presence of LRR9 enhances the polyI:C binding activity and the activation of antiviral immune pathways of ScTLR19. These findings provide a new insight into the mechanisms underlying the different strategies employed by barbel chub and grass carp in response to viral infection and the genetic basis of disease resistance in the hybrid offspring of barbel chub and grass carp. The study is innovative, and the manuscript is well written. I recommend a minor revision, and several issues should be addressed before acceptance.
--Abstract
The importance of TLRs in antiviral immunity of fish needs to be introduced in Abstract.
Response: Toll-like receptors (TLRs) play indispensable roles in antiviral immunity of fish. We have clarified this point in Abstract.
The first letter “Docking Score” should be lowercase.
Response: Thanks. We have modified them according to the reviewer’s suggestion.
--Induction
Line 75: What does “their” stand for?
Response: Thanks. “their” stands for fish TLR19s, and we have clarified this point in the revised manuscript.
The hybrid offspring of barbel chub and grass carp exhibit stronger resistance to grass carp reovirus (GCRV) infection than grass carp. Is there any reported data to support it before?
Response: Thanks. After GCRV infection, the hybrid offspring of barbel chub and grass carp showed survival rate of 96.7%, while the survival rate of grass carp was 41.7%. We have cited the previous study [Liu Q. Ph.D. Thesis. Hunan Agriculture University; Changsha, China: 2014. Studies on Genetic Characteristics and Grass Carp Reovirus Resistance of F1 Hybrids between Grass Carp (Ctenopharyngodon idellus) and Barbel Chub (Squaliobarbus curriculus)] in the revised manuscript.
--Results
Line 108, the word “the” before ScTLR19 should be remove.
Lines 119-120, “similarity of” should be replaced with “amino acid similarity of”.
Lines 129 and 131, TLR19 should be replaced with TLR19s.
Line 152, complete should be deleted.
Line 157-158, please note the singular and plural forms.
Lines 212, 213, 349 and 350, the first letter “Docking” should be lowercase.
Lines 269 and 364, “studies” should be replaced with “study”.
Response: Thanks. We have corrected these English grammar and diction errors according to the reviewer’s suggestion in the revised manuscript.
--Discussion
Lines 343-344, the authors should provide the full names of AGCRV, VHSV, and IHNV.
Response: Thanks. All of them belong to fish reoviruses. We have clarified this point in the revised manuscript.
Line 359, the authors should provide the full name of type I IFN.
Response: Thanks. We have provided the full name of type I IFN in the revised manuscript.
Line 410, homologs should be replaced with homolog.
--Materials and Methods
There should be a space between numbers and °C.
Line 534, 4C?
Response: Thanks. We have corrected these English grammar, formatting and spelling errors in the revised manuscript.
Reviewer 2 Report
Comments and Suggestions for Authors
1. The discussion section needs to be more in-depth
2. The references cited in the article are too old. Please add references from the past five years
3. Please explain clearly why the hybrid offspring have higher disease resistance than the parents
Author Response
Reviewer #2:
1. The discussion section needs to be more in-depth.
Response: Thanks. The hybrid offspring of barbel chub Squaliobarbus curriculus and grass carp Ctenopharyngodon idella exhibit stronger resistance to grass carp reovirus (GCRV) infection than grass carp, and barbel chub is consider the donor of resistance gene resources. Toll-like receptors (TLRs) play indispensable roles in antiviral immunity, while TLR19 is a fish-specific TLR. To date, studies on fish TLR19s are not systematic and in-depth, and most of them are focused on molecular identification and expression level analysis. In this study, the structures and antiviral immune functions of barbel chub TLR19 (ScTLR19) and grass carp TLR19 (CiTLR19) were compared for the first time. Structurally, ScTLR19 has one additional LRR9 in the extracellular region compared to CiTLR19. Functionally, the presence of LRR9 enhances the polyI:C binding activity of ScTLR19 and enables it to mediate antiviral immune signaling pathways different from those of CiTLR19. ScTLR19 can simultaneously regulate the IRF3-TRIF and IRF3-MyD88 signaling pathways, which may allow ScTLR19 to mediate more extensive anti-GCRV immune signals and functional roles, thereby enhancing the GCRV resistance of barbel chub. These findings may provide the key basis for further exploration of antiviral functions of fish TLR19s and the mechanisms underlying the different strategies employed by barbel chub and grass carp in response to GCRV and the genetic basis of GCRV resistance in the hybrid offspring of barbel chub and grass carp. By adding 19 relevant new references, we have tried our best to discuss these findings more deeply and systematically. We have clarified these points in the resubmitted manuscript.
2. The references cited in the article are too old. Please add references from the past five years.
Response: We have done our best to update the reference list so that the proportion of references published in the last five years is 55.73%.
3. Please explain clearly why the hybrid offspring have higher disease resistance than the parents.
Response: Thanks. In fact, our team have obtained the hybrid offspring of grass carp and barbel chub as early as 11 years ago. In our previous study, the hybrid offspring of barbel chub and grass carp showed survival rate of 96.7% after GCRV infection, while the survival rate of grass carp was 41.7%. This previous study also speculated that the recombination of dominant gene resources in the process of hybridization might lead to the higher disease resistance for the hybrid offspring of barbel chub and grass carp. We have cited the previous study [Liu Q. Ph.D. Thesis. Hunan Agriculture University; Changsha, China: 2014. Studies on Genetic Characteristics and Grass Carp Reovirus Resistance of F1 Hybrids between Grass Carp (Ctenopharyngodon idellus) and Barbel Chub (Squaliobarbus curriculus)] in the revised manuscript. However, more scientific explanations should be given by genome-wide systematic comparisons for barbel chub and grass carp in the future.
Reviewer 3 Report
Comments and Suggestions for Authors
Please see the appthed file for comments to authors.

Author Response
Reviewer #3:
The authors demonstrated that the presence of LRR9 enhances the polyI:C binding activity of ScTLR19 and enables it to mediate anti-viral immune signaling pathways different rom those of CiTLR19. Sc TLP19 simultaneously regulate the IRF3-TRIF and IRF3-MyD88 signaling pathway, which may allow ScRLR19 to mediate more extensive anti-GCRV resistance of barbel chub. This is a wellprepared study and well written manuscript containing new findings. Only minor editorial revisions are suggested.
1. Line 53: add (MyD88) after myeloid differentiation factor 88.
2. Line 120: delete % after 68.2.
3. Line 165: barbel chub (Sc), grass carp (Ci), common carp (Cc), channel catfish (Ip), zebrafish (Dr) and Atlantic salmon (Ss).
4. Line 278: delete P>0.05.
5. Line 312: (C. iedella).
6. Line 313: (S. curriculus).
7. Line 429: add (citation) after our laboratory.
8. Line 430: add , city, state before USA.
9. Line 431: add , city, state before USA.
10. Line 476: add city before PA.
11. Lines 509-510: delete , Dalian, China.
12. Lines 510, 515: add , city, state before USA.
13. Lines 516, 517: delete , USA.
14. Line 518: delete % after 70.
15. Line 520: add city, state before USA.
16. Line 522: add Tokyo bfore , Japan.
17. Line 528: add St. Louis, MO, USA.
18. Line 536: delete St. Louis, MO, USA.
19. Line 556: delete , USA after Fisher Scientific.
20. Line 557: delete , USA after Fisher Scietific.
21. Line 560: add city, state before USA.
22. References: titles should not be used large capitals; each word of Journal names should be large capital for the first letter of each word; such as Veterinary Immunology and Immunopathology, Current Opinion in Immunology, International Journal of Biological Macromolecules, Fish & Shellfish Immunology, Molecular Immunology etc: delete : official journal of the International Society for Interferon and Cytokine Research, (New York, N.Y.), (San Diego, Calif.).
Response: We very appreciate the Reviewer #3 for carefully reading our manuscript and providing positive and constructive comments and suggestions. We have carefully checked the whole of manuscript, and improved these English grammar, diction, formatting and bibliographic citation according to the reviewer’s suggestion. The revised text has been marked in red to indicate the changes in the resubmitted manuscript.